# Altered Auditory and Visual Evoked Potentials following Single and Repeated Low-Velocity Head Rotations in 4-Week-Old Swine

**DOI:** 10.3390/biomedicines11071816

**Published:** 2023-06-25

**Authors:** Anna Oeur, William H. Torp, Kristy B. Arbogast, Christina L. Master, Susan S. Margulies

**Affiliations:** 1Wallace H. Coulter Department of Biomedical Engineering, Emory University and Georgia Institute of Technology, Atlanta, GA 30332, USA; anna.oeur@emory.edu (A.O.); wtorp3@gatech.edu (W.H.T.); 2Center for Injury Research and Prevention, Children’s Hospital of Philadelphia, Philadelphia, PA 19146, USA; arbogast@chop.edu (K.B.A.); masterc@chop.edu (C.L.M.); 3Perelman School of Medicine, the University of Pennsylvania, Philadelphia, PA 19104, USA; 4Sports Medicine and Performance Center, Children’s Hospital of Philadelphia, Philadelphia, PA 19104, USA

**Keywords:** brain concussion, auditory evoked potentials, visually evoked potentials, auditory perception, visual perception, traumatic brain injuries, electrodes, swine

## Abstract

Auditory and visually evoked potentials (EP) have the ability to monitor cognitive changes after concussion. In the literature, decreases in EP are commonly reported; however, a subset of studies shows increased cortical activity after injury. We studied auditory and visual EP in 4-week-old female Yorkshire piglets (N = 35) divided into anesthetized sham, and animals subject to single (sRNR) and repeated (rRNR) rapid non-impact head rotations (RNR) in the sagittal direction. Two-tone auditory oddball tasks and a simple white-light visual stimulus were evaluated in piglets pre-injury, and at days 1, 4- and 7 post injury using a 32-electrode net. Traditional EP indices (N1, P2 amplitudes and latencies) were extracted, and a piglet model was used to source-localize the data to estimate brain regions related to auditory and visual processing. In comparison to each group’s pre-injury baselines, auditory Eps and brain activity (but not visual activity) were decreased in sham. In contrast, sRNR had increases in N1 and P2 amplitudes from both stimuli. The rRNR group had decreased visual N1 amplitudes but faster visual P2 latencies. Auditory and visual EPs have different change trajectories after sRNR and rRNR, suggesting that injury biomechanics are an important factor to delineate neurofunctional deficits after concussion.

## 1. Introduction

Sports-related traumatic brain injuries (TBI) are one of the leading causes of emergency department visits [1], with an estimated 1.6–3.8 million sports-related TBI occurring in the United States every year [2]. The International Concussion in Sport Group (CISG) defines a sport-related concussion as a TBI induced by a direct or indirect blow that transmits force to the head, resulting in short-duration impairments that can evolve into longer-lasting signs and symptoms [3]. Forces incurred from a TBI affect the brain at the cellular level, with perturbations disrupting membranes and proteins that inhibit transport of ions across energy channels necessary for homeostasis [4]. This, in turn, impairs neural function and can initiate neuroinflammatory responses that are secondary to the initial insult [5]. Typical cognitive impairments associated with concussion are attention, memory, and information-processing, as well as somatic symptoms such as headaches and dizziness [6]. Auditory and visual dysfunctions are a common finding after closed head injury [7] including concussion [8,9,10] and may disproportionately disadvantage children, as problems with learning, reading, and speech may impede developmental social and scholastic success [11,12,13].

Electroencephalography (EEG) is a promising tool for concussion assessment, as it permits millisecond (ms) measurement of synchronous postsynaptic potentials of cortical pyramidal neurons, with the possibility of detecting neurologic dysfunction [14]. EEG assessments for TBI include continuous measurements of brain activity or those evoked by the presentation of a stimulus, defined as evoked potentials (EP) [15]. The EEG waveform resulting from EPs are a series of positive (P) and negative (N) peaks and troughs that are denoted by approximate time of presentation post-stimuli: P1 (50 ms), N1 (75–140 ms), P2 (150–230 ms), N2 (150–250 ms) and P3 (250–350 ms) [16].

Auditory EPs permit an integrative assessment of the auditory pathway from the cochlea, auditory nerve and brainstem pathways, as well as auditory cortical functions that reflect sound detection and early stimulus-processing in the primary auditory cortex [17]. Visual EPs holistically assess pathways from light stimuli on the retina to the optic nerves, the optic chiasm, thalamus and occipital cortex [18]. Auditory and visual EPs have the potential to provide an indication of functional brain activity and related changes in information processing post-TBI that are specific to the brain structures involved with each respective pathway.

The injured brain has been hypothesized to have lower-amplitude and longer-latency responses compared to a healthy brain, reflecting a decreased capacity for information-processing and slower transmission speeds [16,19]. In the literature, there have been studies that support this hypothesis, those that partially support (in amplitude but not latency or vice versa), and even studies that report no differences between concussed and healthy groups. It is also noteworthy that there seems to be a pattern in the literature showing groups with a history of concussion having alternative cognitive processes compared to a no-injury group. To highlight some examples, symptomatic and asymptomatic concussed adults had reduced N1 amplitudes in comparison to a control group; however, there were no differences in latency [20]. Vander Werff and Rieger [8] reported no differences in P1, N1, and P2 between controls and adults with long-term concussion (up to 18 months); however, P3 amplitudes were reduced for the injured group.

In a study involving junior ice hockey players, athletes with a history of concussion (3+) had significantly longer P3 latencies than a no concussion group [21]. More recently, Bennys et al. [22] showed that athletes with a history of concussion (1+ in the last 3 years) had decreased P300 amplitudes and a trend towards longer latencies (in comparison to no injury) from auditory oddball tasks.

Interestingly, some studies report contrary findings where amplitudes are greater and latencies are increased. For example, concussed athletes at 4 years post-TBI had increased N2 and P3 amplitudes, in addition to longer latencies, in comparison to healthy controls [23]. In a separate study, male ice hockey players were subject to a two tone-auditory oddball test within 24 h of concussion and similarly reported increased amplitudes (N100, P300, N400) and decreased latencies in comparison to baseline values [24]. One hypothesis for the increase in cortical activity after TBI posits that recovery from injury could result in a compensatory mechanism by which an increase in neural activity is required to meet the same executive functioning demands as the non-injured brain [23]. It is thought that, after injury, new neural networks (and combinations of neural networks) can engage in different temporal and spatial patterns affecting EP attributes recorded from the scalp [25].

The variability in the timepoints studied post-TBI (acute versus chronic), and the history of concussion per subject (total number and time between injuries) across studies are contribute to the mixed EP findings in the literature. Other factors include heterogeneity in the mechanisms of injury [26], as the mechanical parameters governing the loading conditions in a head injury event, such as velocity and direction of motion, are key causal factors in the observed patterns of neural trauma [27]. In addition, age and cortical maturation have an effect on EP amplitudes and latencies throughout development in childhood [28,29] and adulthood [30], contributing to varying effects of concussion on these measures across the lifespan [31].

Pre-clinical animal models of TBI provide an idealized platform to allow for biomechanical control over the head trauma load and direction; the timepoints studied post-TBI, total number and timing of multiple traumas, as well as animal age and brain maturation to better isolate their effects on brain injury [32,33]. Animal models provide an opportunity to systematically assess the subtleties of TBI and improve our understanding of the auditory and visual impairments related to structurual and functional deficits post-concussion [34,35]. In comparison to other animal models (monkey, dog, rodents), the rapid post-natal development of pigs after birth makes this species a suitable model across a number of different fields, including the skeletal and neuromuscular, pulmonary and cardiovascular, central nervous system and gastrointestinal system [36]. A 4–14-week-old piglet approximates a young child of roughly 2–12 years old [37]. A 4-week-old swine is an established model of pediatric TBI, with neuroanatomical structures (gyri and sulci) and gray and white matter distributions that are similar to the developing brain [38,39] and are important biomechanical characteristics to model brain movement within the skull [40]. The spectrum of diffuse axonal injuries, a form of TBI including concussion, was achieved in our large animal model using a rapid non-impact head rotation (RNR) device and employed in studies from our laboratory examining neurobehavioral deficits, histopathology, and drug efficacy in piglets of different ages [32,41,42,43,44,45].

This study utilizes methods established in our laboratory for measuring evoked potentials in healthy 4-week-old swine, in addition to studying the large-animal model under single or repeated head loads using an RNR device. Sixteen healthy animals presenting with a passive two-tone auditory oddball test were used; infrequent target tones produced greater N1 amplitudes for frontal electrodes and produced consistent day-to-day responses [46]. In a separate set of healthy animals (N = 11), cognitive activity from auditory and visual stimuli were compared using traditional evoked potential measures and cortical activations estimated from source localization techniques. In healthy animals, N1 amplitudes were greater from auditory stimuli in comparison to visual stimuli. P2 amplitudes were greater from visual stimuli and latencies (N1 and P2) were faster for visual stimuli than auditory stimuli. Patterns of cortical activation showed that visual stimulation had greater levels of early (50 ms) activity than auditory stimulation; however, at 85 ms, auditory had greater left-temporal activations. At 110 ms, visual stimulation had greater activity in the left and right occipital regions [47]. The objective of this study was to examine the role of single and repeated head rotations on auditory and visually evoked potentials to better describe the effects of head biomechanics and loading patterns on neurocognitive deficits. Derived from this prior work, our hypotheses for the current study are as follows: (1) there will be a ‘between experimental group’ effect where RNR will significantly reduce EP indices and cortical activation compared to sham; (2) within each experimental group, there will be a ‘day effect’ where cognitive processing is decreased at various timepoints after RNR or anesthesia compared to pre-injury baselines; and (3) there will be a ‘stimulus effect’ where the patterns of reduced cognitive activity are unique to auditory stimuli and are different from visual stimuli.

## 2. Methods

To study the effects of concussion on auditory and visual processing, we employed the 4-week-old swine pediatric TBI model subject to prescribed controlled mild head rotations. Auditory and visually evoked potentials were captured using methods published previously in healthy piglets [46,47] on each animal before head rotations, to establish a pre-injury baseline, and again at 1, 4, and 7 days after to examine the acute time-course of mild TBI on stimulus processing. Other neurofunctional measures were also collected using this experimental injury paradigm and include piglet gait and pupillary light reflexes, using previously published methods [48,49].

### 2.1. Animal Subjects

Thirty-five 4-week-old female Yorkshire piglets were allocated into three experimental groups: sham (N = 10), single RNR (N = 12), and repeated RNR (N = 13). Awake subjects were fitted with a 32-electrode EEG net and evaluated using two-tone auditory oddball tests at baseline and multiple days after rapid non-impact head rotation (RNR) or an anesthesia-only event (sham). A subset of these animals was also evaluated using a simple white-light visually evoked potential: sham (N = 5), single RNR (N = 6), multiple RNR (N = 9). Animals were socially housed in cages on a 12 h light and 12 h dark cycle, and were freely permitted food (LabDiet 5080, St. Louis, MO, USA) and water. EEG measurements were taken in a separate behaviour test room where auditory and visual EP measurements were taken in the awake animal while gently restrained in a sling. Animals were acclimated to the sling and head gear (EEG net or stretchable nylon) for at least two 30 min sessions prior to the first study day. All subjects that were included for analysis survived to the end of study. All animal procedures were approved by Emory University’s Institutional Animal Care and Use Committee (IACUC).

### 2.2. Rapid Non-Impact Rotational Injury (RNR)

On the injury day (day 0), each injured and sham piglet was sedated with an intramuscular injection (Ketamine:4 mg/kg, Xylazine: 2 mg/kg, and Midazolam: 0.2 mg/kg) and underwent anesthesia via inhalation of 1.5% isoflurane using a fitted snout mask. While vitals were monitored and body temperature was maintained, a lack of response to a mild toe pinch confirmed the appropriate depth of anesthesia; then, the RNR was delivered. Sham animals received anesthesia only, without RNR. Animals receiving a head rotation were secured to a HYGE device (HYGE Inc., Kittaning, PA, USA) via a bite plate. The bite plate was attached to a linkage system that transfers the linear motion of the pneumatic actuator to a rapid head rotation of the snout and head [32,41]. Angular transducers were mounted onto the linkage system to measure the angular velocity of the system (ARS-06, ATA Engineering, Inc., Herndon, VA or ARS Pro, DTS Inc., Seal Beach, CA, USA). The HYGE device is capable of RNR at levels consistent with the spectrum of diffuse axonal injury pathologies [50,51,52]. Piglets in the single and repeated RNR groups experienced sagittal rotation on day 0 (Figure 1). The levels of head rotations prescribed to each experimental animal group was computationally scaled from soccer participants instrumented with head-impact sensors [53]. The load levels were scaled from 267 headers in high-school soccer players, primarily causing sagittal motions of the head [53]. The human kinematic data were input into a finite element model of the human brain and maximum axonal strain (MAS) was estimated for each of the 267 impacts [53,54]. The 50th (medium) and 90th (high) percentile MAS values were extracted from the human header data and scaled using a piglet finite-element model to determine the corresponding peak angular velocity and angular accelerations associated with the 50th (medium) and 90th (high) levels. The single RNR group received a single ‘high header’ and the repeated RNR group experienced one ‘high header’ load followed by four ‘medium header’ loads. The target ‘high header’ rotations scaled to the pig were 104 ± 2.36 rad/s and 37.8 ± 6.06 krad/s^2^ and ‘medium header’ rotations were 61.2 ± 2.02 rad/s and 15.0 ± 1.72 krad/s^2^. For the repeated-RNR group, rotations were delivered 8.4 min apart (±1.1 min). The 8 min interval between impacts for the repeated RNR group was determined from the same high-school soccer heading data, where girls and boys received 4 and 6 impacts per h, respectively, where impacts were spaced 8 min apart for both sexes. Further details on the determination of piglet head rotation magnitudes are described in [48]. EEG measurements were taken on a pre-injury day (D-1), one (D1), four (D4) and seven days (D7) post-injury (Figure 1). The Institutional Animal Care and Use Committees (IACUC) at Emory University approved all animal procedures conducted in this study.

### 2.3. Electroencephalography Measurements

Non-invasive EEG data were collected using scalp electrodes embedded in a custom 32-channel EGI HyrdoCel piglet electrode net at 1000 Hz using a Net Amps 400 amplifier (Electric Geodesics Inc., EGI, Eugene, OR, USA). Prior to application on the piglets, the net was soaked in baby shampoo (5 mL), potassium chloride (10 mL) and water (1 L) solution. The net was then placed on the piglet’s head and electrical impedance was checked and maintained below 1 kΩ [55]. EEG data acquisition was accomplished using a MacBook Pro laptop with Netstation (Version 5.0, Electric Geodesics Inc., EGI, Eugene, OR, USA) and synchronized to auditory and visual stimuli using E’Prime on a PC computer (Version 2.0, Psychology Software Tools, Inc., Pittsburgh, PA, USA). The auditory oddball train consisted of a 100-tone clicktrain comprising 70 standard tones (800 Hz) and 30 target tones (1000 Hz) played in random order (Figure 1A). For visual trains, 30 white-light flashes were presented using a 7-inch LCD screen (Figure 1B). Auditory sounds were presented to the centre of the piglets’ head, attempting to stimulate both ears equally; however, for visual stimuli, the light stimuli was presented to the left eye only. Stimuli were presented in the same manner as in our study with healthy piglets [47]. Each animal received 6 trials of auditory trains, and the animals who were studied for visually evoked potentials also received 6 trials of visual trains [47].

EEG data were pre-processed in Netstation Tools (Electric Geodesics Inc., EGI, Eugene, OR, USA) and included band-pass filtering from 0.1 to 30 Hz, segmentation of evoked potential into 300 ms epochs that incorporate a 50 ms pre-stimulus baseline and a 250 ms post-stimulus response period. Bad channels were replaced if greater than 200 uV via interpolations of two nearby electrodes [47]. Data were post-processed in EEGlab Version 14.12 [56] and Matlab Version R2018b (The Mathworks, Inc., Natick, MA, USA) [57] and included baseline correction, waveform averaging and independent component analysis to remove noise artefacts and eye movements [58].

EEG waveforms were averaged per animal, per day and per stimulus type (auditory-target, -standard, and visual). N1 and P2 peak amplitudes and latencies were extracted from the averaged waveforms from each electrode. N1 and P2 pertain to the first negative and second positive peak following stimulus presentation and are the most consistent attributes of the evoked potential for piglets [47]. Peak data from electrodes 1, 2, 3, 4, 17, and 27 were averaged together to represent the response at the front of the head. Similarly, electrodes were averaged together to represent activity in the left (5, 11, 13, 15, 23) and right (6, 12, 14, 16, 24) temporal regions. Values were removed if the peak was non-negative and non-positive for N1 and P2, respectively.

### 2.4. Source Localization

EEG waveforms were source-localized using a finite-element model (FEM) of the piglet head and brain to estimate the electrical activity distribution in the brain in sham and RNR piglets. Details regarding the development of this model for source localization in healthy piglets are described elsewhere [47]. Briefly, the model was derived from magnetic resonance images (MRI) of an infant piglet and scaled to the 4-week-old pig [59]. The model comprises 1.6 million tetrahedral elements making up the scalp, skull, brain, ventricles, and eyes [59,60]. The 32-electrode array was imported on the scalp of the model [61]; however, eight electrodes (9, 10, 19, 20, 21, 22, 25, 26) were not included as they lay in a region behind the ears. Conductivity values were assigned for the scalp (0.4 S/m), skull (0.03 S/m), cerebral spinal fluid (1.79 S/m), brain (0.5 S/m) and eyes (1.5 S/m) [59]. Standardized low-resolution brain electromagnetic tomography algorithms (sLORETA) were employed to calculate the inverse solution [62,63]. Mean current density was extracted from five regions of interest (frontal, left and right temporal, and left and right occipital) at three timepoints post-stimuli (50 ms, 85 ms, and 110 ms). These timepoints were selected to capture the cortical activity surrounding the N1 amplitude, which was found in our previous study to have the largest amplitude response from auditory and visual stimuli [47].

### 2.5. Statistics

Three-way repeated-measure ANOVAs were run per electrode region (frontal, left and right temporal) to evaluate the effect of day (repeated measures), experimental group (sham, sRNR, rRNR), and stimulus (standard, target, and visual) on N1 and P2 amplitudes and latencies. Statistical analyses for source localization were run in a similar manner, employing 3-way repeated measures ANOVAs to evaluate the effect of day (repeated measures), experimental group (sham, sRNR, rRNR), and stimulus (standard, target, and visual) on current density. This analysis was repeated 15 times, to stratify according to timepoint (50 ms, 85 ms, 110 ms) and brain region (frontal, left and right temporal, left and right occipital regions). Post hoc analyses employed one-way ANOVAs with Bonferroni corrections. All statistics were conducted using IBM SPSS (Version 25, Armonk, NY, USA: IBM Corp.) and significance was accepted at *p* < 0.05.

## 3. Results

### 3.1. Overview

As an overview, the mean and standard deviation of the angular velocities and angular accelerations corresponding to medium and high load levels for the single-RNR and repeated-RNR groups are reported in Table 1. The measured head kinematics were consistent with our target load levels, as determined from scaling. Exemplar current density patterns for each group at 85 and 110 ms are illustrated in Figure 2 and Figure 3. The results for source localization are shown in Figure 2 and Figure 3. Evoked potential results are shown in Figure 4, presented by stimulus type because our findings from healthy animals showed that visual stimuli produced larger responses in the occipital areas and more auditory stimuli in the temporal regions [47]. Figure 5, Figure 6 and Figure 7 illustrate 50, 85, and 110 ms current density results, respectively for an exemplar sRNR animal across study day and stimulus type.

Comparisons of EP and current density measurements that were significantly different from each experimental group’s pre-injury values and contrary to those found in healthy animals from a previous study [47] are reported in this section. Table 2 presents a summary of the main findings for each experimental group, highlighting changes from pre-injury baselines.

### 3.2. Overall Group Differences

The findings examining experimental group comparisons are presented next. For reliability, if there were significant differences between experimental groups at pre-injury for an extracted EP parameter or current density for a region, further group comparisons on subsequent days were not examined. Figure 2 presents example patterns of cortical activity from source localization analysis at day 4 for a single sham, sRNR, and rRNR animal presented with a standard tone and a visual stimulus depicted at 85 ms. We will discuss the results in each region. In the frontal region, statistically significant results were found, where auditory processing was decreased for RNR groups in comparison to sham. Specifically, on day 4, rRNR (1.117 ± 0.203 µV) had decreased target P2 amplitudes compared to sham (2.672 ± 0.321 µV) and sRNR (0.035 ± 0.005 µV/mm^2^) had lower activations from standard tones compared to sham (0.046 ± 0.009 µV/mm^2^) at 85 ms.

**Figure 2 biomedicines-11-01816-f002:**
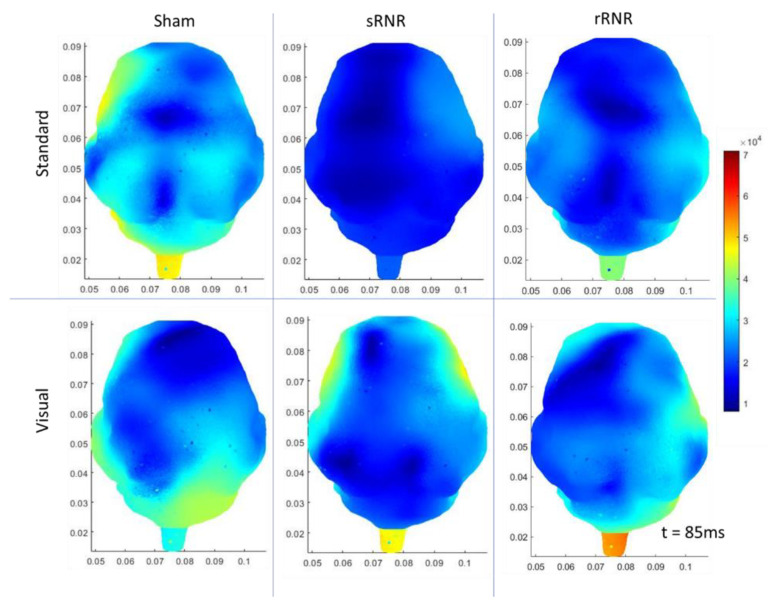
Exemplar current density distributions (85 ms) for sham (**left**), sRNR (**middle**) and rRNR (**right**) for standard tones (**top** row) and visual stimuli (**bottom** row) taken at day 4, illustrating unique cortical activity patterns across experimental group and stimulus type. Between-group comparisons found that sRNR showed significantly decreased frontal activity compared to sham for standard tones. rRNR had significantly decreased left-temporal activity than sRNR for visual stimuli, but was not different than sham.

In the left temporal region, there were pre-injury differences for current density between sRNR and sham (110 ms) for visual stimuli N1 latencies; however, these were not different at pre-injury and, therefore, sRNR had significantly longer visual N1 latencies (62.067 ± 2.811 ms) than sham (49.400 ± 3.443 ms) at day 1. No pre-injury differences were observed between sRNR and rRNR; therefore, on day 4, sRNR had greater left temporal activations than rRNR at 85 ms (sRNR: 0.071 ± 0.012 µV/mm^2^, rRNR: (0.031 ± 0.006 µV/mm^2^) and 110 ms (sRNR: 0.070 ± 0.012 µV/mm^2^, rRNR: 0.035 ± 0.007 µV/mm^2^) post visual stimulus (Figure 3).

Regarding pre-injury, in the right temporal regions, standard P2 amplitudes were significantly different between rRNR and sRNR and visual P2 latencies were different between sRNR and sham. P2 amplitudes were not significantly different between rRNR and sham; therefore, contrary trends were found between groups in this region on day 1, where rRNR (2.695 ± 0.360 µV) had greater standard P2 amplitudes than sham (0.830 ± 0.360 µV). In addition, sRNR had greater visual activations (85 ms) than sham (sRNR: 0.050 ± 0.008, sham: 0.022 ± 0.005 µV/mm^2^). Further, sRNR (2.824 ± 0.265 µV) and rRNR (2.915 ± 0.342 µV) had greater target P2 amplitudes than sham (1.364 ± 0.42 µV) on day 4, with sRNR (2.538 ± 0.277 µV) remaining significantly greater than sham (0.684 ± 0.358 µV) on day 7.

In the left occipital region, pre-injury current densities at 110 ms were significantly different between sRNR and rRNR for standard tones only. Early cortical activity in the left occipital region (50 ms) showed greater activations from target tones for sRNR (0.027 ± 0.004 µV/mm^2^) than rRNR (0.024 ± 0.007 µV/mm^2^) on day 1. Interestingly on day 4, sham (0.027 ± 0.003 µV/mm^2^) had greater activations than rRNR (0.022 ± 0.004 µV/mm^2^) from standard tones; however, this was reversed on day 7, where rRNR (0.024 ± 0.004 µV/mm^2^) was greater than sham (0.015 ± 0.003 µV/mm^2^). Additionally, sRNR (0.102 ± 0.024 µV/mm^2^) had greater left occipital activations (85 ms) from visual stimuli than sham on day 7 (0.025 ± 0.006 µV/mm^2^). In the right occipital region, there were pre-injury differences between sRNR and sham, and sRNR and rRNR, with no other significant group comparisons on other days or stimuli.

**Figure 3 biomedicines-11-01816-f003:**
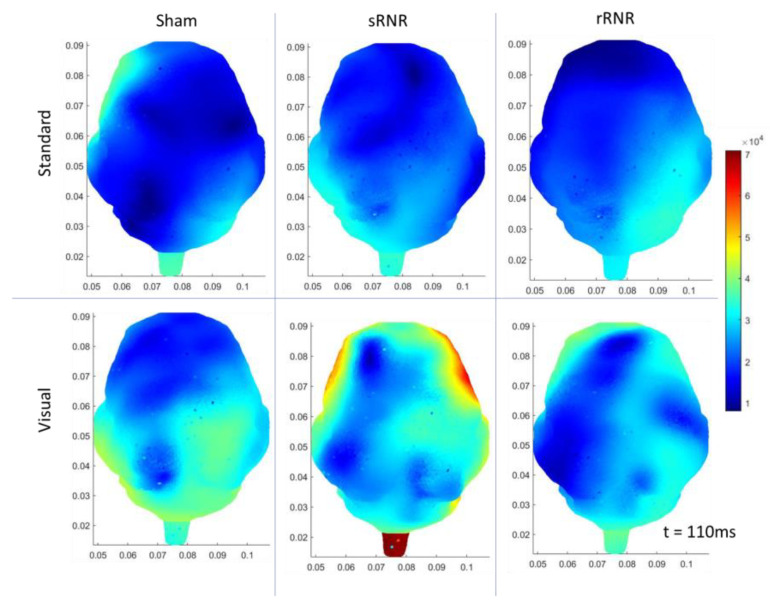
Exemplar current density distributions (110 ms) for sham (**left**), sRNR (**middle**) and rRNR (**right**) for standard tones (**top** row) and visual stimuli (**bottom** row) taken at day 4. Between-group comparisons found that sRNR had significantly increased left-temporal activity compared to rRNR for visual stimuli.

### 3.3. Within Group Differences

Differences found within each experimental group (sham, sRNR, and rRNR) are presented below.

#### 3.3.1. Sham–Anaesthesia Only

In sham animals, frontal N1 (−4.848 ± 0.848 µV) and P2 (1.388 ± 0.678 µV) amplitudes from target tones were significantly decreased at day 1 (N1: −2.711 ± 0.598 µV) and at day 7 (P2: 0.476 ± 0.915 µV) compared to baseline pre-anesthesia levels (Figure 4, top panel). In our previous study, examining the auditory oddball paradigm in healthy piglets, there was no effect of day, where standard and target tones produced similar responses on subsequent days of testing [46]. Similarly, frontal activations determined from source localization were decreased at day 1 (0.042 ± 0.008 µV/mm^2^) and 7 (0.033 ± 0.004 µV/mm^2^) in comparison to pre-anesthesia (0.057 ± 0.009 µV/mm^2^) as a result of standard tones (85 ms time point). At the 85 ms timepoint, the left temporal region showed decreased activations at day 1 for both standard (pre: 0.073 ± 0.014, day 1: 0.059 ± 0.010 µV/mm^2^) and target tones (pre: 0.084 ± 0.011, day 1: 0.065 ± 0.010 µV/mm^2^). We conclude that anesthesia influenced auditory responses for many metrics on day 1, with some persisting to day 7. No significant changes (from pre-anesthesia) were found for visual stimuli.

**Figure 4 biomedicines-11-01816-f004:**
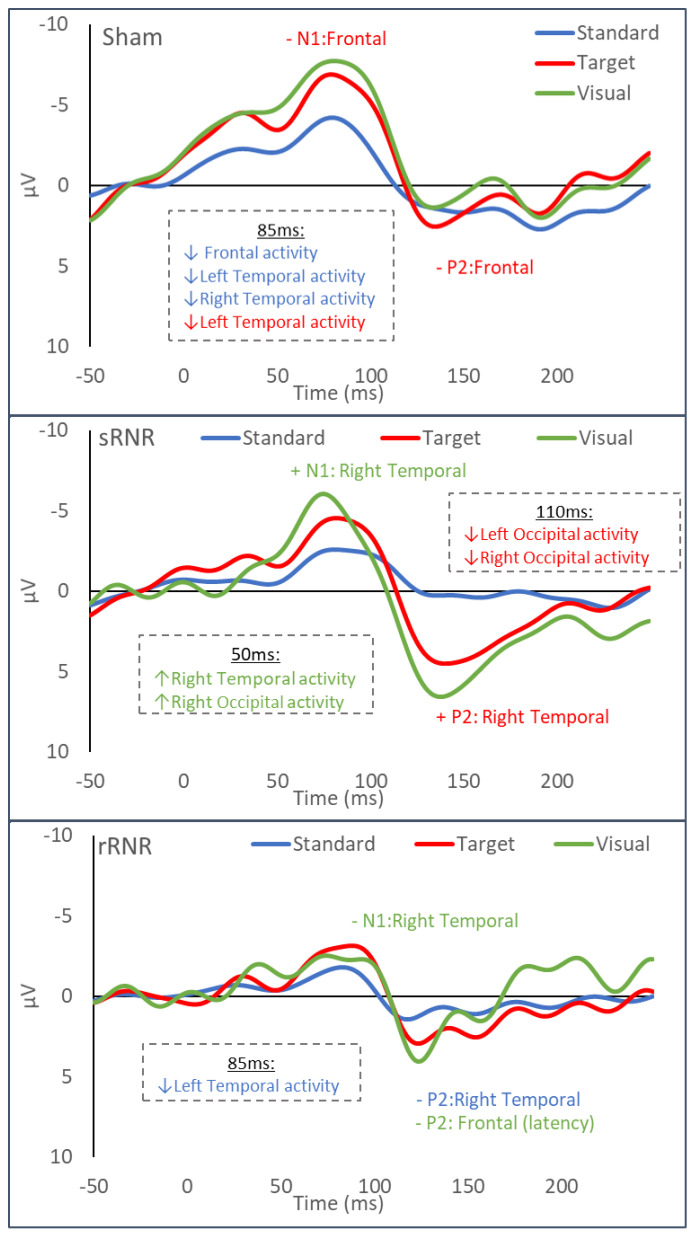
Single-subject exemplar evoked potential (EP) waveforms from channel 17 taken on day 1, summarizing group findings for sham (**top**), sRNR (**middle**), and rRNR (**bottom**), where blue represents auditory standard tones, red target tones, and green visual. Significant EP findings are denoted by ‘−‘ and ‘+’, indicating decreases and increases from pre-injury for each brain region. All EP findings pertain to significant amplitude changes, except for P2 frontal latency in rRNR, as indicated in parentheses. For each plot, significant current density findings are summarized in square (dotted) boxes for each specified timepoint (50, 85, and 110 ms) and brain region.

#### 3.3.2. sRNR-Single

For the sRNR group, auditory target tones produced a significant increase in P2 amplitudes in the right temporal region from pre-injury (1.878 ± 0.245 µV) to day 7 (2.538 ± 0.277 µV). Similarly, visual stimuli (Figure 4) produced an increase in right temporal N1 amplitudes from pre-injury (−0.622 ± 0.429 µV) to day 1 (−1.827 ± 0.368 µV). Source localization results from visual stimuli show that, at 50 ms (Figure 5), the right temporal had significantly greater activations at day 7 (0.101 ± 0.011 µV/mm^2^) than both pre-injury (0.050 ± 0.009 µV/mm^2^) and day 1 (0.056 ± 0.007 µV/mm^2^) timepoints. At 50 ms, the right occipital regions also showed a significant increase in visual stimuli activations at day 7 (0.109 ± 0.017 µV/mm^2^) compared to pre-injury levels (0.069 ± 0.011 µV/mm^2^). This region also increased at day 4 (0.100 ± 0.014 µV/mm^2^), above the pre-injury baseline. At 50 ms, for auditory stimuli, there were no injury-associated effects in any region. Figure 6 shows current density patterns at 85 ms; however, no significant changes were found between days and stimuli for sRNR at this timepoint. At 110 ms, auditory target tones produced significantly decreased activations at day 1 in the left occipital (pre: 0.069 ± 0.011, day 1: 0.043 ± 0.004 µV/mm^2^) and right occipital (pre: 0.064 ± 0.005, day 1: 0.049 ± 0.006 µV/mm^2^) regions (Figure 7). At 110 msec, for auditory standard tones and visual stimuli, there were no injury-associated effects in any region.

**Figure 5 biomedicines-11-01816-f005:**
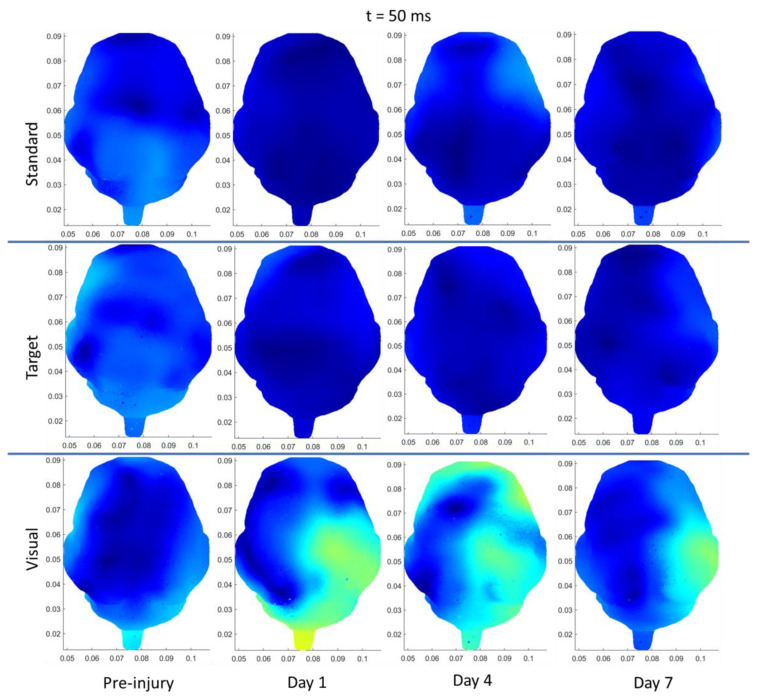
Depiction of source localization analysis at 50 ms post-standard (**top** row), target (**middle** row), and (**bottom** row) visual stimuli for one exemplar animal from the sRNR group across days studied (preinjury, day 1, 4, and 7).

**Figure 6 biomedicines-11-01816-f006:**
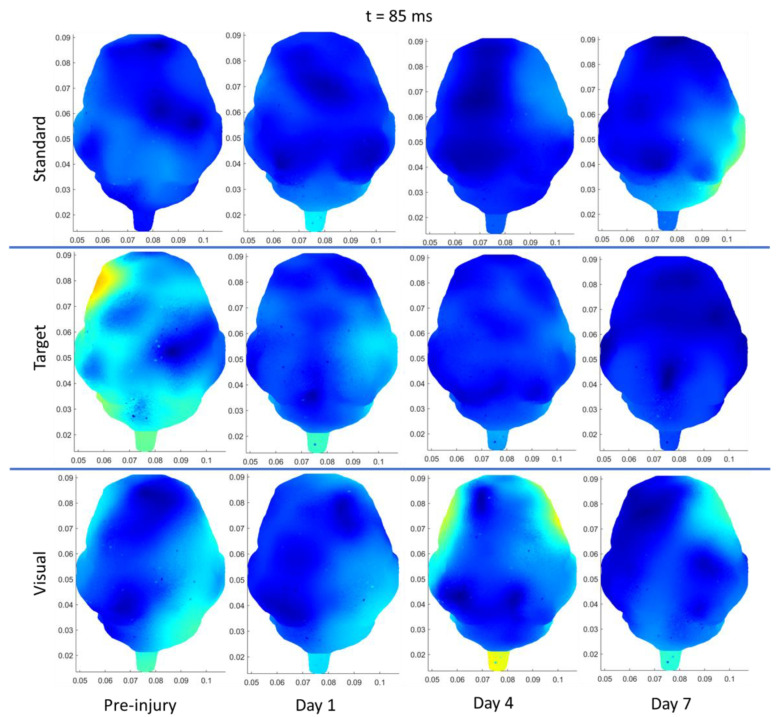
Depiction of source localization analysis at 85 ms post-standard (**top** row), target (**middle** row), and (**bottom** row) visual stimuli for one exemplar animal from the sRNR group across days studied (preinjury, day 1, 4, and 7).

**Figure 7 biomedicines-11-01816-f007:**
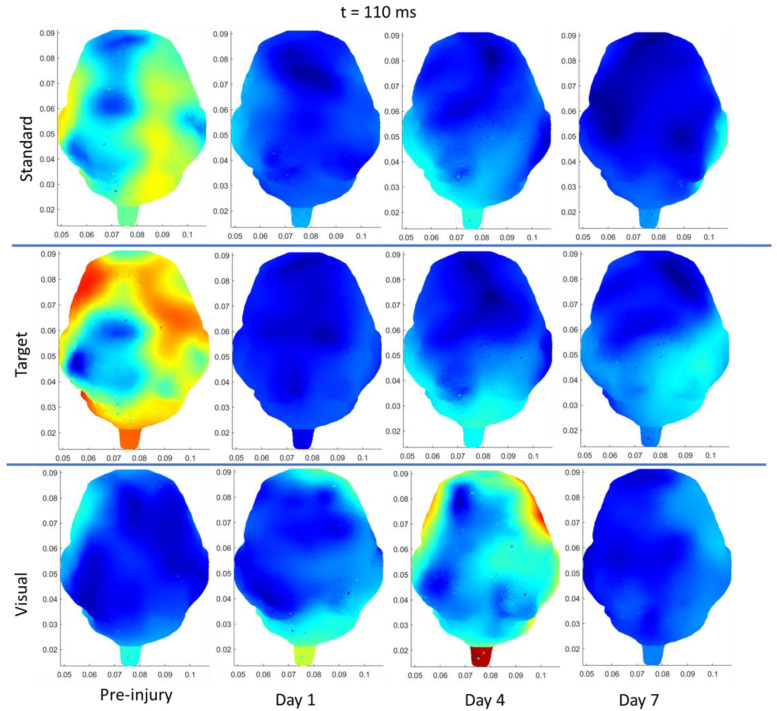
Depiction of source localization analysis at 110 ms post-standard (**top** row), target (**middle** row), and (**bottom** row) visual stimuli for one exemplar animal from the sRNR group across days studied (preinjury, day 1, 4, and 7).

#### 3.3.3. rRNR–Repeated

In rRNR animals, right-temporal N1 and P2 amplitudes were found to be significantly decreased on day 1 for visual (pre: −2.121 ± 0.350, day 1: −1.240 ± 0.350 µV) and day 4 for auditory standard tone (pre: 3.593 ± 0.515 µV, day 4: 1.626 ± 0.431 µV) stimuli, respectively. In addition, frontal P2 latencies were faster at day 4 (111.4 ± 5.497 ms), in comparison to pre-injury (144.0 ± 5.747 ms) from visual stimuli. Source localization for auditory stimuli results yielded significant findings at 85 ms, where standard tones had decreased activations from pre-injury (0.072 ± 0.006 µV/mm^2^) compared to day 7 (0.048 ± 0.004 µV/mm^2^) in the left temporal region (Figure 3). However, no differences between pre- and post-injury were found at 50 or 110 ms for any stimuli on any day.

Different trajectories of auditory and visual processing changes were noted for sham, sRNR and rRNR groups. In anesthesia-only sham, auditory processing was affected, but not visual processing. In the sRNR group, there were mixed findings, where visual and auditory stimuli resulted in increased and decreased responses, respectively. In the rRNR group, the magnitude of the response (amplitude or current density) had a tendency to decrease. Across all groups, the frontal region emerges as the most vulnerable to auditory deficits, as evidenced by decreased cortical activity; however, visual deficits were observed in the left temporal regions. Interestingly, we found increased activity in the right temporal and right occipital regions that were not specific to any stimulus modality.

## 4. Discussion

This study examined acute changes (within 7 days) in auditory and visually evoked potentials in a large 4-week-old swine model of TBI under single or repeated head-loads using an RNR device. This current study is an extension of previous work that established methods for measuring and modelling auditory and visually evoked potentials in healthy swine and applied these methods to an experimental cohort subject to RNR. A table summarizing significant EP changes in auditory and visual processing for sham (anesthesia-only) and RNR is displayed in Table 2. In the acute post-injury phase, anaesthesia and RNR were found to have an effect on 4-week-old swines’ EP amplitudes and latencies and cortical activations in comparison to pre-injury. In sham, only auditory stimuli were found to be significantly decreased post-anaesthesia on days 1 and 7 in comparison to pre-injury values (Figure 4, top panel). This finding is consistent with the human literature, where auditory processing was suppressed after anesthesia [64]. Visually evoked potential remained unaffected after anaesthesia in piglets, as no significant differences were found on any post-anaesthesia days (1, 4 or 7) following pre-injury.

Interesting findings were observed for the sRNR group, where increased activity (N1 and current density) was observed for visual stimuli in the right temporal and right occipital regions, however auditory-target processing yielded the opposite results, with an increase in P2 amplitudes in the right temporal region but decreased current densities in the left and right occipital regions (Figure 4, middle panel). In healthy piglets, auditory processing localized to the temporal region and visual processing to the occipital regions [47]. In this study, these regions show alternative activity patterns after experimental RNR. The rRNR group yielded different patterns to the sRNR group, where visual stimuli produced decreased N1 amplitudes but faster visual P2 latencies (Figure 4, bottom panel). The different patterns of findings between RNR groups suggest that rRNR is not necessarily a more severe form of sRNR, as this group did not simply reflect greater magnitude deficits within the same parameters or within the same region. sRNR was the only experimental group that showed increased EP responses, while rRNR only decreased responses, in addition to there being fewer significant comparisons than were found in the sRNR group (Table 2). Differences in activation patterns observed between RNR groups further highlight the importance of distinguishing between single and repeated head rotations, as the human literature demonstrates a trend toward the worst outcomes for those suffering from a history of concussion versus no injury or a single concussion [19]. The ‘within group’ findings for each RNR combination were unique to the mode of stimuli and suggest that the visual and auditory pathways are unequivocally affected by each head-loading paradigm. Taken together, these findings demonstrate the importance of injury biomechanics, particularly head-loading patterns, that create differing trajectories in neurofunctional deficits. A mixture of head injury mechanisms is one factor that likely contributes to the incongruent findings in the human literature, and should be considered when interpreting trends in neurocognitive deficits and symptoms.

Comparisons between groups revealed that both sRNR (current density) and rRNR (P2 amplitude) were decreased in comparison to sham; a finding consistent with the human literature, where auditory information processing is decreased after mild TBI in clinical and athletic populations [26,65,66]. Furthermore, the sRNR group was found to have slowed visual processing (left temporal) in comparison to sham (N1 latency), and rRNR had lower visual activity than sRNR in this region. In previous studies, visual processing in response to a working memory task requiring an active button-press had smaller N350 and P300 amplitudes for patients with mTBI in comparison to healthy controls [67], as well as decreased activity in the dorsolateral prefrontal cortex from functional magnetic resonance imaging (fMRI) [68]. Furthermore, patients with a history of concussion reported decreased P3 amplitudes, as it has been hypothesized that the brain has an increased vulnerability to subsequent concussions resulting in worse outcomes [19,22].

Contrary trends were observed in the right temporal and left occipital regions in our animals, as there were many instances in which increased activity (P2 amplitudes and current densities) were observed for either the sRNR or rRNR groups in comparison to sham. These findings were not specific to a type of stimulus. Increased activity in these regions after RNR suggests that the brain may involve more areas (and at a greater magnitude) to process the same amount of information. Enhanced auditory processing (increased amplitude and decreased latency) was observed in an evaluation of N100, P300, and N400 from a two tone-auditory oddball task in junior ice hockey players taken at pre-season, after a concussion, and multiple timepoints within- and post-season [24]. In a separate study examining varsity athletes, enhanced auditory N2 and P3 and increased electrical activity (from source localization modelling) were reported in athletes with a history of concussion in comparison with a non-concussed group [23]. Similarly, in a visual Go/No-Go task employing an emotional cue (a spider for a threat-related condition or a flower for a neutral condition), enhanced N2-P3 was observed for individuals with a concussion in comparison to control group. The authors postulated that increased attention and cortical activity in response to threat conditions may be related to inefficient control of the emotional response in visual processing in the concussed group [69]. It has been theorized that increased cortical activity after TBI (in contrasted to suppressed activity) can resilt from a breakdown of neural networks, resulting in the inefficient processing of information, requiring more areas to become active to compensate [23,25].

This study examined the effects of single and repeated mild head rotations on auditory and visually evoked potentials in the 4-week-old swine model permitting systematic control of the loading conditions causing transient neurofunctional deficits common to concussion. A limitation of human studies is that the biomechanics causing head injury are often diverse, while factors such as direction, velocity, surface stiffness, are not controlled for in the analysis [27]. These biomechanical factors govern the conditions of energy transfer from an event to cause injury to the head and brain tissues. Despite careful control of the loading conditions in this study, we observed trends where auditory and visual processing not only decreased, but also increased after head rotations. While head rotations were completed in the sagittal plane by applying loads evenly across both hemispheres, it is possible that the dysfunctions were unequal for the auditory and visual pathways because each involves different brain structures. We acknowledge that the piglet brain is much smaller in size than the pediatric brain and the neural axis of the pig is parallel to the ground whereas humans’ is perpendicular [40], further limiting the direct translation of findings this study to humans. Furthermore, we noted that patterns of increases or decreases in auditory and visual processing were specific to the examined brain region, thus supporting the notion of whole-brain analysis to capture the full picture of deficits or over-compensation mechanisms after TBI. We did observe an effect of anaesthesia on decreased auditory processing in sham animals; however, we conducted within-group comparisons in our attempt to delineate the effects of anaesthesia versus RNR on outcomes to better isolate and understand the effects of RNR. Further limitations in this study include employing passive auditory and visual tasks that were suitable for piglets, while the majority of tasks in the human literature employ active tasks requiring a response or more complex and cognitively demanding tasks. We studied RNR in a single direction and only in female pigs. We hypothesize that the magnitude and direction (increase or decrease) of auditory and visual processing would change if a different direction, i.e., coronal loading, was employed, as the biomechanical loads would disproportionally affect these pathways as the head would moved from medial to lateral instead of from anterior to posterior directions. Male pigs may have different auditory and visual processing trends after RNR as it has been demonstrated in the human literature that males with concussion have a greater N1 suppression after the visual presentation of human faces of different emotions in comparison to females with concussions [70]. Lastly, our findings were limited to the 35 animal subjects studied, and it is possible that, if more animals were included, comparisons that were found to be not significant may become significant.

## 5. Conclusions

In summary, the trajectory of alterations in auditory and visually evoked potentials were increased after single RNR but decreased after repeated RNR. This suggests that the injury processes affecting cortical activity for the rRNR group may be different than those for the sRNR group since rRNR did not simply reflect greater changes in the same direction (increased activity). The frontal region seems to be most vulnerable to auditory deficits, as reflected by decreased cortical activity, and visual deficits were found in the left temporal regions. Interestingly, we found increased activity in the right temporal and right occipital regions that were not specific to any stimulus modality. Auditory and visual EPs have different change trajectories after sRNR and rRNR, suggesting that injury biomechanics are important factors when delineating the patterns of neurofunctional deficits after concussion. Auditory and visual stimuli evaluate separate and specific neural pathways of the brain; however, future work should include other stimulus modalities, such as motor or sensory pathways, to examine the integrative brain functionality after TBI and provide a complete understanding of the functional changes after concussion.

## Figures and Tables

**Figure 1 biomedicines-11-01816-f001:**
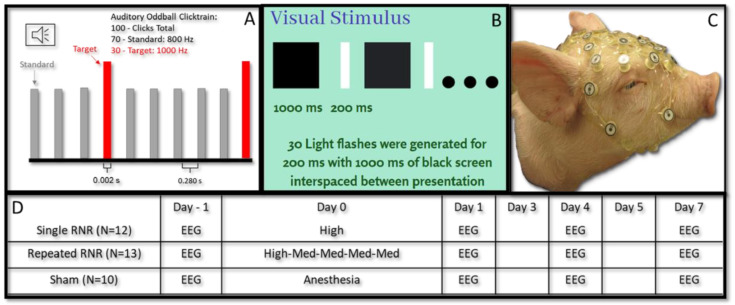
Summary auditory oddball paradigm with 30 target (1000 Hz) tones (0.002 s) randomly played in between standard (800 Hz) tones with an interstimulus interval of 0.280 s (**A**). Simple white light visual stimulus (**B**). Custom 32-electrode EEG net on piglet (**C**). Experimental timeline for sRNR, rRNR, and sham animals for EEG data collection and injury/anesthesia (**D**) at pre-injury (pre), injury (D0), day 1 (D1), day 4 (D4) and day 7 (D7). RNR = rapid nonimpact rotation.

**Table 1 biomedicines-11-01816-t001:** Mean ± standard error of angular velocity and acceleration loading levels for single- and repeated-RNR groups.

	Load Level	Angular Velocity (rad/s)	Angular Acceleration (rad/s^2^)
sRNR	High	104 ± 0.495	36,900 ± 1120
rRNR	Medium	61.2 ± 0.182	14,900 ± 175
	High	104 ± 0.414	38,300 ± 533

**Table 2 biomedicines-11-01816-t002:** Summary table of significant changes in auditory and visual EP findings for sham and RNR piglets.

	Region	Stimulus	Significant Change from Pre-Injury
Sham	Frontal	Auditory–Standard	Decrease (85 ms, current density)
	Frontal	Auditory–Target	Decrease (N1 and P2 amplitudes)
	Left Temporal	Auditory–Standard	Decrease (85 ms, current density)
	Left Temporal	Auditory –Target	Decrease (85 ms, current density)
	Right Temporal	Auditory–Standard	Decrease (85 ms, current density)
sRNR	Right Temporal	Visual	Increase (N1 amplitude)
	Right Temporal	Auditory–Target	Increase (P2 amplitude)
	Right Temporal	Visual	Increase (50 ms, current density)
	Right Occipital	Visual	Increase (50 ms, current density)
	Left Occipital	Auditory–Target	Decrease (110 ms, current density)
	Right Occipital	Auditory–Target	Decrease (110 ms, current density)
rRNR	Right Temporal	Visual	Decrease (N1 amplitude)
	Right Temporal	Auditory–Standard	Decrease (P2 amplitude)
	Frontal	Visual	Decrease (P2 latency) *
	Left Temporal	Auditory–Standard	Decrease (85 ms, current density)

NB: ‘*’ denoting a decrease in latency means P2 latencies were faster rRNR.

## Data Availability

Not applicable.

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
