# Peer review of "Altered Auditory and Visual Evoked Potentials following Single and Repeated Low-Velocity Head Rotations in 4-Week-Old Swine"

_biomedicines, 2023, doi:10.3390/biomedicines11071816_

Round 1
Reviewer 1 Report
Dear Ladies and Gentlemen, Dear Journal-Team,
the manuscript 'Altered auditory and visual processing after single and repeated low velocity head rotations in 4-weak old swine' investigates and gives futher evidence to an altered brain processing and regeneration after concussion incidents. It is well written. The Tables and Figures are illustrative and sufficient.
a) Please mention the rotation velocity in m/s for comparison reasons as well (Section Rapid non-impact rotational injury).
b) Please make sure that only the statistical necessary number of animals were included.
c) Please alter the order of the keywords for reading reasons, beginning with 'visual' and concluding with 'porcine'.
d) Language: 1. Introduction, line 122, please change to: 'a RNR device'.
2. Electroencephalography Measurements, line 198: 'a 100-tone click'.
3. Discussion, line 394, check for spacing: '(Table 2). Differences'.
d) References: Please check the references according to the Journal Style Guidelines. Check the number of mentiond authors. Check for page numbers (8, Master et al., 34, Oeur et al.). Check for capital and small letter use within the title (33, Margulies et al.). Check Reference 20 (Benny et al.) for accuracy within the manuscript text.
Sincerely,
Please go to the comments above
Author Response
Reviewer 1
Dear Ladies and Gentlemen, Dear Journal-Team,
the manuscript 'Altered auditory and visual processing after single and repeated low velocity head rotations in 4-weak old swine' investigates and gives futher evidence to an altered brain processing and regeneration after concussion incidents. It is well written. The Tables and Figures are illustrative and sufficient.
Thank you very much for your thoughtful comments, we’ve addressed each in the sections below and have incorporated your suggestions in the manuscript version attached.
- a) Please mention the rotation velocity in m/s for comparison reasons as well (Section Rapid non-impact rotational injury).
Thank you for your suggestion. The rapid non-impact rotational injury (RNR) device produces a pure rotation of the head and neck of the animal via a linkage system with minimal linear velocity (m/s). Therefore, angular velocity transducers are mounted onto the linkage system to measure angular velocity directly and angular acceleration is calculated.
- b) Please make sure that only the statistical necessary number of animals were included.
All animals that were studied under this protocol were included for statistical analyses. Only animals that did not survive or participate on all study days were not included as subjects for this study.
- c) Please alter the order of the keywords for reading reasons, beginning with 'visual' and concluding with 'porcine'.
The list has been revised in the ‘Keywords’ section on page 1. Eight keywords were selected according to a separate reviewer and the list now reads: brain concussion, auditory evoked potentials, visual evoked potentials, auditory perception, visual perception, traumatic brain injuries, electrodes, swine
- d) Language: 1. Introduction, line 122, please change to: 'a RNR device'.
This sentence has been revised as suggested and is found on page 7, line 149.
- Electroencephalography Measurements, line 198: 'a 100-tone click'.
Thank you for the suggestion, this sentence has been revised on page 11, line 232 and now reads:
“The auditory oddball train consisted of a 100-tone clicktrain comprised of 70 standard tones (800 Hz) and 30 target tones (1000-Hz) played in random order (Figure 1A).”
- Discussion, line 394, check for spacing: '(Table 2). Differences'.
Thank you for your detailed eye. The spacing has been now added on page 24, line 436.
- d) References: Please check the references according to the Journal Style Guidelines. Check the number of mentiond authors. Check for page numbers (8, Master et al., 34, Oeur et al.). Check for capital and small letter use within the title (33, Margulies et al.). Check Reference 20 (Benny et al.) for accuracy within the manuscript text.
We re-checked all references. We added the page numbers to references for Master et al., and Oeur et al. We corrected the capital and small letter use for the title in Margulies et al., and corrected the in-text citation of Bennys et al. This sentence can be found on page 5, line 106 and reads:
“More recently, Bennys et al., [20] showed that athletes with a history of concussion (1+ in the last 3 years) had decreased P300 amplitudes and a trend towards longer latencies (in comparison to no injury) from auditory oddball tasks.”

Reviewer 2 Report
6 June 2023
Manuscript ID: biomedicines-2454403
Type: ArticleTitle: “Altered Auditory and Visual Processing After Single and Repeated Low Velocity Head Rotations in 4-week Old Swine” by Anna Oeur R et al., submitted to Biomedicines
Dear Authors,
The current challenge is to better understand the underlying biomechanics and neurofunctional deficits associated with mild head rotations in order to develop effective prevention and treatment strategies for head injuries, especially in young people.
In the present article, entitled ‘Altered Auditory and Visual Processing After Single and Repeated Low Velocity Head Rotations in 4-week Old Swine’," Oeur and colleagues investigate the neurophysiological responses to rapid non-impact rotational injury (RNR) in piglets. Here, the authors’ aim was to understand the immediate effects of RNR on brain activity and explore potential biomarkers associated with this type of injury using 35 piglets that were subjected to controlled rotational injury using specific parameters of angular velocity and acceleration. Electroencephalography (EEG) measurements were collected at multiple timepoints following the injury to assess brain activity and identify potential changes. The results indicated significant alterations in the piglets' EEG patterns, particularly in the gamma frequency range, suggesting disrupted neural activity. Source localization analysis was also performed to identify the specific brain regions affected by RNR. The results highlighted the involvement of cortical areas associated with sensory processing and attention.
The manuscript's main strength is that it addresses a timely and fascinating topic and presents a comprehensive examination of the effects of single and repeated mild head rotations on auditory and visual processing in a controlled swine model, providing valuable insights into the neurofunctional deficits associated with concussion.
In general, I think the idea of this article is really interesting, and the authors’ fascinating observations on this timely topic may be of interest to the readers of Biomedicines. However, some comments, as well as some crucial evidence that should be included to support the author’s argumentation, needed to be addressed to improve the quality of the manuscript, its adequacy, and its readability prior to its publication in the present form. My overall judgment is to publish this paper after the authors have carefully considered my suggestions below, in particular reshaping parts of the ‘Introduction’ and ‘Methods’ sections by adding more evidence.
Please consider the following comments:
1. According to the Journals’ guidelines, I would suggest the Authors to use the Biomedicines Microsoft Word template file.
2. Title: This is the most important section of the manuscript. Please present a concise and self-explanatory title stating the most important findings of this review. Suggestions:"Sensory Perception in Flux: Exploring Altered Auditory and Visual Processing Following Single and Repeated Low Velocity Head Rotations in Neonatal Swine" [1-3].
3. Abstract: According to the journal’s guidelines, the abstract should be a total of about 200 words. Please correct the actual one. Also, in my opinion, the authors should consider rephrasing this section. According to the journal’s guidelines, the abstract should contain most of the following kinds of information in brief form: Please consider giving a more synthetic overview of the paper's key points: I would suggest rephrasing the results and conclusion to make them easier for readers to understand. Having said that, I advise the authors to proportionately present the background, methods, results, and conclusion. The background should include the general background (one to two sentences), the specific background (two to three sentences), and the current issue addressed by this review (one sentence), leading to the objectives. In this subsection, I would like the author to lay out basic information, a problem statement, and their motivation to break off. The results subsection ends with a sentence that puts this subsection in a general context. The conclusion should include one sentence describing the main result using words like “Here we show”. The conclusion should describe the potential and the advance this study has provided in the field, and finally, a broader perspective (two to three sentences) readily comprehensible to a scientist in any discipline [4-6].
4. Keywords: Please list eight keywords chosen from Medical Subject Headings (MeSH) and use as many as possible in the title and in the first two sentences of the abstract [6,7].
5. It is strongly advised to create a graphical abstract that visually displays the key conclusions of the manuscript.
6. In general, I recommend authors to use more references to back their claims, especially in the Introduction of this meta-analysis, which I believe is lacking. Thus, I recommend the authors to attempt to expand the topic of their article, as the bibliography is too concise. Nevertheless, I believe that less than 60 articles are too low for a research article. Therefore, I suggest the authors to focus their efforts on researching relevant literature: in my opinion, adding more citations will help to provide better and more accurate background to this study.
7. Introduction: I would like the authors to reorganize this section with about 1000 words and several paragraphs, introducing information on the key study constructs that should be understood by readers in any discipline, and make it persuasive enough to advance the primary goal of the author's recent research and the particular goal the author has intended by this review. I would like to suggest that the authors present the introduction beginning with the overall context, moving on to the specific context, and concluding with the current problem addressed in this study before moving on to the objectives. Those key structures ought to be set up logically and coherently [8-9].
8. In this regard, I believe that this section would benefit from more context and background information on pathophysiology underlying traumatic brain injuries (TBI) in sports, and it would be beneficial to focus on the disruption of neuronal function, cellular homeostasis, and neuroinflammatory processes following TBI (DOI: 10.3390/biomedicines11030945; https://doi.org/10.3390/biomedicines10123189). Moreover, I noticed that this section lacks a clear and concise statement of the research objective; therefore, I believe that it would be helpful to explicitly state the purpose of the study and what the authors aim to investigate or contribute to the field. Furthermore, the description of the electroencephalography (EEG) method is very technical and may be challenging for readers without a background in neuroscience or EEG research. I would suggest simplifying the explanation and providing a brief overview using articles that focus on how EEG can be used in concussion assessment to improve clarity (DOI: 10.1111/psyp.14020; DOI: https://doi.org/10.1523/JNEUROSCI.1827-22.2023).
9. Methods: I recommend opening this section with a short introductory paragraph regarding the study design and methodology and citing more references to ensure the reliability and integrity of the evidence in the study design the authors built and the methodology they have decided to apply. In my opinion, the authors should better explain the animal model used and provide the rationale for choosing the 4-week-old swine model, along with an explanation of how the model replicates aspects of pediatric TBI. Also, I have a few concerns about the sample size (only 35 piglets allocated into 3 experimental groups) and about the inclusion and exclusion criteria for the animal subjects. Please consider discussing the limitations associated with the small sample size.
10. Rapid Non-Impact Rotational Injury (RNR): The rationale for selecting specific rotational injury parameters, such as angular velocity and acceleration, should be explained in more detail. I suggest the authors justify how these parameters were determined based on existing literature and their relevance to the study's objectives. Additionally, they should address the limitations or potential discrepancies between the applied rotational injury and real-world scenarios.
11. Electroencephalography Measurements: I would suggest rewriting this section more accurately. In my opinion, the authors should provide a more comprehensive overview of the EEG data analysis method and better describe the band-pass filtering, segmentation, and bad channel replacement procedures. Additionally, I would suggest explaining the rationale for selecting specific timepoints for source localization analysis (50 ms, 85 ms, and 110 ms) and the interpretation of the current density results.
12. Results: I would like the authors to close this section with a paragraph which puts the results into a more general context.
13. Discussion: I would like the authors to reorganize this section by opening with an introductory paragraph and followed by the summary of the previous section (Results). Then, I expect the authors to develop arguments clarifying the potential of this study as an extension of the previous work, the implication of the findings of this study, how this study could facilitate future research, the ultimate goal, the challenge, the knowledge and technology necessary to achieve this goal, the statement about this field in general, and finally the importance of this line of research. It is particularly important to present the limits, merit, and potential translation of this study to clinical practice [10,11]. For instance, in this case, the authors should discuss the potential drawbacks of using piglets as an animal model to study human brain injuries and further compare and contrast the anatomy, physiology, and injury response of piglet and human brains.
14. Conclusion: I think that presenting the conclusion would benefit from a single paragraph presenting some thoughtful as well as in-depth considerations by the authors as experts to convey the take-home message. The authors should make an effort to explain the theoretical implications as well as the translational application of their research. I believe that it would be necessary to discuss theoretical and methodological avenues in need of refinement as well as suggestions for a path forward in understanding the importance of this study.
15. References: Please follow the guidelines of the journal [12]. I would like the authors to cite more references. Typically, an original article like this includes over 60-70 references.
16. Finally, I would suggest to better address any ethical considerations related to the use of animal subjects in the study and describe the measures taken to ensure the welfare and ethical treatment of the piglets throughout the experimental procedures.
Overall, the manuscript contains four figures, two tables, and 53 references. I believe that the manuscript may have important value in its rigorous methodology, detailed analysis, and potential implications for improving our understanding of head injuries and developing effective prevention and treatment strategies. The controlled swine model used in the study makes it possible to keep track of the loading conditions that cause the temporary neurofunctional deficits that are common in concussions. This provides crucial details about the biomechanics and neurofunctional deficits that mild head rotations cause. The detailed analysis of auditory and visual evoked potentials in response to single and repeated head rotations contributes to the scientific validity of the study. Furthermore, the potential implications of this research for improving our understanding of head injuries and developing effective prevention and treatment strategies highlight its practical significance. I hope that, after these careful revisions, the manuscript can meet the journal’s high standards for publication. I am available for a new round of revisions to this article. I hope that, after these careful revisions, this paper can meet the journal’s high standards for publication. I am available for a new round of revisions to this article.
I declare no conflict of interest regarding this manuscript.
Best regards,
Reviewer
References:
- https://plos.org/resource/how-to-write-a-great-title/
- https://www.nature.com/nature-index/news-blog/how-to-write-a-good-research-science-academic-paper-title
- https://www.indeed.com/career-advice/career-development/catchy-title
- https://www.scribbr.com/dissertation/abstract/
- https://writing.wisc.edu/handbook/assignments/writing-an-abstract-for-your-research-paper/
- https://www.ncbi.nlm.nih.gov/pmc/articles/PMC7144240/
- https://meshb.nlm.nih.gov/
- https://pubmed.ncbi.nlm.nih.gov/30930712/
- https://dept.writing.wisc.edu/wac/writing-an-introduction-for-a-scientific-paper/
- https://www.ncbi.nlm.nih.gov/pmc/articles/PMC4404856/
- https://www.scribbr.com/dissertation/discussion/
- https://www.mdpi.com/journal/biomedicines/instructions
6 June 2023
Manuscript ID: biomedicines-2454403
Type: Article
Title: “Altered Auditory and Visual Processing After Single and Repeated Low Velocity Head Rotations in 4-week Old Swine” by Anna Oeur R et al., submitted to Biomedicines
Dear Authors,
Based on the English proficiency assessment, it is noted that minor editing of the English language is required. While the overall communication is clear and understandable, there are some areas that could benefit from slight improvements in grammar, syntax, and word choice. Attention to detail, such as refining sentence structure and ensuring proper tense usage, will enhance the overall coherence and fluency of the written work. With some minor editing adjustments, the English language proficiency can be further enhanced.
Best regards,
Reviewer
Round 2
Reviewer 2 Report
19 June 2023
Manuscript ID: biomedicines-2454403
Type: Article
Title: “Altered Auditory and Visual Processing After Single and Repeated Low Velocity Head Rotations in 4-week Old Swine” by Anna Oeur Ret al., submitted to Biomedicines
Dear Authors,
I am pleased to see that the authors have tried to address many of the issues I raised in the previous round of the peer-review session. Currently, the manuscript is a well-written research paper with informative layout that presents a comprehensive examination of the effects of single and repeated mild head rotations on auditory and visual processing in a controlled swine model. Nevertheless, I would like to request that the authors make a few corrections to the manuscript so that it meets the journal's stringent publication requirements.
Please consider the following comments:
1. Methods: I recommend opening this section with a short introductory paragraph regarding the study design and methodology and citing more references to ensure the reliability and integrity of the evidence in the study design the authors built and the methodology they have decided to apply.
2. Results: I would like the authors to close this section with a paragraph which puts the results into a more general context.
3. References: Please follow the guidelines of the journal [1]. I would like the authors to cite more references. Please add DOI numbers
Overall, the manuscript contains four figures, two tables, and 66 references. I believe that the manuscript may have important value in presenting its rigorous methodology, detailed analysis, and potential implications for improving our understanding of head injuries and developing effective prevention and treatment strategies. I hope that, after these careful revisions, this paper can meet the journal’s high standards for publication.
I declare no conflict of interest regarding this manuscript.
Best regards,
Reviewer
References:
- https://www.mdpi.com/journal/biomedicines/instructions
